# Realizing Video Summarization from the Path of Language-based Semantic Understanding

## Abstract

The recent development of Video-based Large Language Models (VideoLLMs), has significantly advanced video summarization by aligning video features—and, in some cases, audio features—with Large Language Models (LLMs). Each of these VideoLLMs possesses unique strengths and weaknesses. Many recent methods have required extensive fine-tuning to overcome the limitations of these models, which can be resource-intensive. In this work, we observe that the strengths of one VideoLLM can complement the weaknesses of another. Leveraging this insight, we propose a novel video summarization framework inspired by the Mixture of Experts (MoE) paradigm, which operates as an inference-time algorithm without requiring any form of fine-tuning. Our approach integrates multiple VideoLLMs to generate comprehensive and coherent textual summaries. It effectively combines visual and audio content, provides detailed background descriptions, and excels at identifying keyframes, which enables more semantically meaningful retrieval compared to traditional computer vision approaches that rely solely on visual information, all without the need for additional fine-tuning. Moreover, the resulting summaries enhance performance in downstream tasks such as summary video generation, either through keyframe selection or in combination with text-to-image models. Our language-driven approach offers a semantically rich alternative to conventional methods and provides flexibility to incorporate newer VideoLLMs, enhancing adaptability and performance in video summarization tasks.

**Input Video**

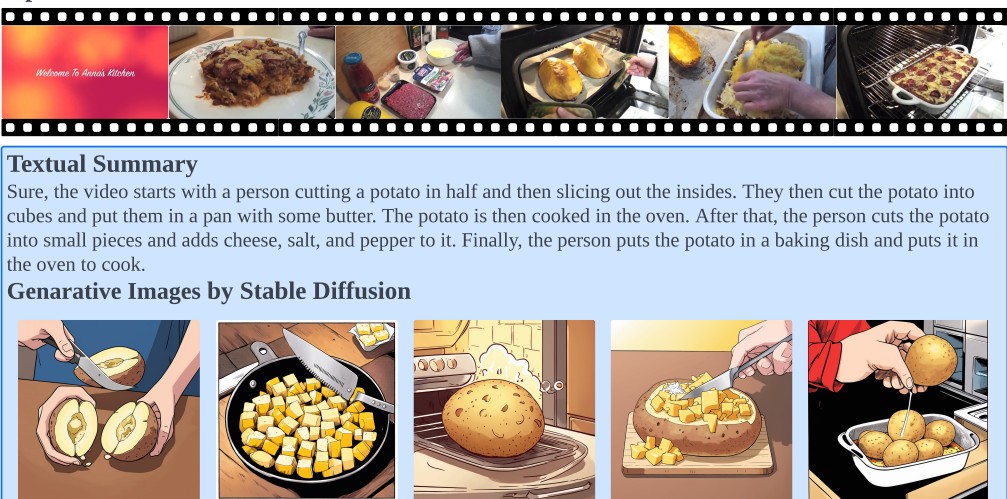

**Textual Summary**
Sure, the video starts with a person cutting a potato in half and then slicing out the insides. They then cut the potato into cubes and put them in a pan with some butter. The potato is then cooked in the oven. After that, the person cuts the potato into small pieces and adds cheese, salt, and pepper to it. Finally, the person puts the potato in a baking dish and puts it in the oven to cook.

**Genarative Images by Stable Diffusion**

Figure 1: Visualization of two of our extended applications on HowTo100M (Miech et al., 2019). The "Input Video" refers to the keyframes selected from the original video based on our textual summary, and pairing with our textual summary, we can simulate **visual manual generation**. The images generated by Stable Diffusion 3 (Esser et al., 2024) simulate **privacy-preserving content generation**.

# 1 INTRODUCTION

In recent day, the proliferation of video content across various platforms has led to an overwhelming amount of information, making it challenging for users to efficiently access and digest the key information. As a result, video summarization has emerged as a crucial task, enabling the efficient extraction of key segments from lengthy videos. The goal of video summarization is to condense extensive video content into textual summaries, short video clips, or a collection of representative images. Given the vast amount of video data generated daily, effective summarization not only enhances user experience by reducing the time required to access essential information but also supports efficient content management and retrieval across platforms. Additionally, video summarization has significant applications in areas such as surveillance, education, entertainment, and multimedia indexing, making it a vital tool for navigating and leveraging the vast expanse of video data available today.

The success of Visual Language Models (VLMs) (Liu et al., 2024; Wang et al., 2023b; Alayrac et al., 2022) has paved the way for the development of Video LLMs, such as VideoLLaMA (Zhang et al., 2023), VideoChat (Li et al., 2023a), and VideoLLaVA (Lin et al., 2023a). These VideoLLMs leverage human-annotated data for instruction tuning, and they propose different methods to align video features with the LLM feature space. Each model exhibits distinct strengths: for example, PG-Video-LLaVA (Munasinghe et al., 2023) demonstrates pixel-grounded capabilities for capturing detailed scenes, Video-LLaMA adopts a multi-branch cross-modal framework that incorporates audio information in addition to video content, and LLaMA-VID (Li et al., 2023b) excels in capturing background scene details. However, despite these strengths, existing VideoLLMs have inherent shortcomings and lack coherent methods to address them. For instance, Video-LLaVA and LLaMA-VID are unable to retrieve audio signals, while Video-LLaMA lacks the grounding abilities required for retrieving fine-grained details. Additionally, LLMs within these VideoLLMs often suffer from hallucination issues. To overcome these limitations, previous approaches typically resort to fine-tuning or retraining models, which can be computationally expensive. Our observation, however, suggests that the limitations of one VideoLLM can often be mitigated by the strengths of another. This leads us to ask: ***What if we could utilize existing VideoLLMs collaboratively, instead of resorting to costly fine-tuning or retraining of a new model?***

In this work, we draw inspiration from the Mixture of Experts (MoE) (Shen et al., 2023; Lin et al., 2024) paradigm, which is designed to enhance performance in processing large and complex tasks by leveraging multiple expert sub-models. Specifically, our approach employs multiple VideoLLMs for video summarization, integrating the concept of LLM cooperation to combine the outputs from these video "experts" through our proposed inference-time algorithm. This method allows us to address the limitations of individual VideoLLMs by compensating with the strengths of other expert VideoLLMs. Furthermore, since our framework does not require fine-tuning or retraining, it can seamlessly adapt to incorporate new or updated VideoLLMs as additional expert models.

Overall, we propose a novel video summarization method that follows a unique path of language-based semantic understanding. By proposing an inference-time algorithm, we can generate comprehensive textual summaries that capture not only visual content but also audio information, providing detailed descriptions of background scenes to offer users a more holistic view of the original videos. Additionally, with our comprehensive textual summaries, we can perform various downstream video summarization tasks, such as identifying keyframes and generating images and videos, thereby surpassing the capabilities of existing VideoLLMs.

Our main contributions can be summarized as follows:

- We propose an inference-time algorithm that leverages the capabilities of LLMs to combine the output summaries of multiple VideoLLMs into a single, coherent, and unbiased summary. This approach provides more detailed and comprehensive information, enhancing the overall quality of video summarization.

- Additionally, our comprehensive and coherent summaries enhance keyframe retrieval with a simple keyframe selection algorithm, surpassing the performance of existing approaches.

- Our proposed method is both flexible and general. The components of our framework can be easily replaced with more powerful models. Moreover, it is general enough to support

extended video applications that can leverage our intermediate outputs, such as textual summaries and keyframes.

## 2 RELATED WORK

**Large Language Models.** Large Language Models (LLMs) have revolutionized the field of natural language processing (NLP) and artificial general intelligence (AGI) with their exceptional capabilities in language generation, in-context learning, and reasoning. The historical evolution of these models began with foundational architectures such as BERT (Devlin, 2018), GPT-2 (Radford et al., 2019), and T5 (Raffel et al., 2020), which set the stage for subsequent advancements. The introduction of GPT-3 (Brown, 2020), with its 175 billion parameters, marked a significant breakthrough, showcasing remarkable performance across a wide spectrum of language tasks. This progress spurred the development of an array of other influential LLMs, including Megatron-Turing NLG (Smith et al., 2022), Chinchilla (Hoffmann et al., 2022), PaLM (Chowdhery et al., 2023), OPT (Zhang et al., 2022), BLOOM (Le Scao et al., 2023), LLaMA (Touvron et al., 2023), MOSS (Sun et al., 2024), and GLM (Zeng et al., 2022). These models, characterized by their scale and open-source availability, have become invaluable for both training large models and fine-tuning them for specific applications.

**Visual Language Models.** With the emergence of LLMs, recent works (Liu et al., 2024; Wang et al., 2023b; Alayrac et al., 2022) have increasingly explored their use in processing visual inputs, giving rise to Visual Language Models (VLMs). The central idea behind this line of work is to align visual features with the textual features of LLMs by utilizing a common framework. This framework typically involves a pretrained visual encoder to extract visual features, a projection layer to map these visual representations into the text latent space of LLMs, and the pretrained LLM to generate responses, thereby enabling the powerful capabilities of LLMs to be applied to vision tasks. Video-based Large Language Models (VideoLLMs) extend the capabilities of VLMs by incorporating temporal and/or audio features, allowing for richer video-language understanding through human-video dialogue interactions. For instance, methods such as VideoChatGPT (Maaz et al., 2024) and Valley (Luo et al., 2023) use pooling over visual tokens to obtain compact visual representations. VideoChat (Li et al., 2023a) employs pretrained video foundation models and Q-Former (Zhang et al., 2024) from BLIP-2 (Li et al., 2022) to aggregate video representations. Video-LLaMA (Zhang et al., 2023) introduces a Video Q-Former and an Audio Q-Former for multimodal video comprehension. Furthermore, MovieChat (Song et al., 2024) proposes an advanced memory management mechanism for reasoning over extended video content.

**LLM Evaluator.** The field of Natural Language Generation (NLG) evaluation has evolved considerably over the years, launching from traditional metrics to more advanced methodologies, particularly with the advent of LLMs. Early metrics, such as ROUGE (Lin, 2004) and BLEU (Papineni et al., 2002), have been foundational in assessing the quality of generated text by comparing it to reference texts based on n-gram overlap. However, these methods have limitations in capturing deeper semantic nuances. To address this, embedding-based metrics like BERTScore (Zhang et al., 2019) were introduced, measuring the semantic similarity between texts using word and sentence embeddings. With the rise of LLMs, evaluation methods have further advanced. LLM-based evaluators, such as GPTScore (Fu et al., 2023), G-Eval (Liu et al., 2023a), and UniEval (Zhong et al., 2022), leverage the comprehensive understanding and generation capabilities of LLMs to provide deeper insights into NLG quality. Recognizing the inherent limitations of these early approaches, subsequent studies concentrated on enhancing factual accuracy (Min et al., 2023), ensuring interpretability (Lu et al., 2024), reducing position bias (Wang et al., 2023a), and aligning evaluation more closely with human judgment standards (Liu et al., 2023b). These efforts represent a significant shift toward more robust and human-aligned evaluation methods in NLG.

## 3 METHODOLOGY

Our holistic video summarization framework, illustrated in Figure 2, is composed of three key modules. In Section 3.1, we introduce two components that utilize VideoLLM "experts" to produce textual summaries within the video summarization module. In Section 3.2, we present our keyframe

Figure 2: An overview of our framework. Our approach consists of three main modules: **(1) Video Summarization**, which constructs coherent textual summaries by leveraging multiple existing VideoLLMs and our proposed inference-time algorithm; **(2) Keyframe Retrieval**, which identifies key moments based on our textual summary using a simple keyframe selection algorithm; and **(3) Extended Applications**, which utilize our informative textual summaries and keyframes to address real-world tasks beyond traditional video summarization.

retrieval module, which details how video frames and textual summaries are projected into a joint embedding space to identify relevant keyframes. Finally, in Section 3.3, we explore the extended applications of our framework, demonstrating how the textual summaries and corresponding keyframes can be used for real-world applications.

## 3.1 VIDEO SUMMARIZATION

We perform inference on the given input video using multiple VideoLLMs. To fully leverage the capabilities of these models, we design and employ prompts specifically tailored to the architecture of each VideoLLM. This approach results in four unique summaries, each capturing different aspects of the input video and reflecting the strengths of each model.

### 3.1.1 DENOISE-AND-COOPERATE

There are two primary challenges in utilizing the generated summaries from these VideoLLMs. First, each VideoLLM exhibits varying degrees of the "hallucination" issue, which can mislead users and make us difficult to identify the inaccuracies specific to each model. Second, effectively integrating and combining the "strengths" of each model from the resulting summaries is a complex task. To address these challenges, we propose the following strategies:

**Filter Outliers.** We propose two outlier filtering strategies to remove the summaries that deviate from the others, that is, from the four distinct summaries generated in the previous step, we identify and exclude the summary that exhibits the lowest similarity to the other three, considering it an outlier. For the first strategy, we reference the scoring method from Open-Sora[1] to evaluate the summaries generated by each VideoLLM. By calculating the matching score between each summary and the middle frame of the video, we identify and remove the summary with the lowest score. As for the second strategy, we aim to enhance the video-text alignment between the generated summaries and the input video, our implementation is outlined in Algorithm 1. This involves calculating the average CLIP score across the summaries and discarding the one with the lowest score.

**Cooperate.** After filtering outliers, we leverage the capabilities of state-of-the-art LLMs, to combine the remaining summaries into a single coherent paragraph. We propose three distinct strategies for this synthesis: Merge, Find Common Ground, and Select.

• Merge: This strategy integrates all information from the VideoLLM summaries into a comprehensive single summary, capturing the full spectrum of details provided by each model. The resulting summary aims to be inclusive and detailed.

---

[1]https://github.com/hpcaitech/Open-Sora/tree/main/tools/scoring (last accessed: 2024/09)

- Find Common Ground: This approach focuses on extracting and consolidating only the common elements across all VideoLLM summaries. The process produces a coherent summary that emphasizes the most consistent and reliable information, potentially reducing noise and inconsistencies.

- Select: This strategy chooses the summary that achieves the highest score based on our evaluation metric in the outlier filtering stage. We find this approach particularly effective for certain video types, such as instructional videos in datasets like HowTo100M (Miech et al., 2019).

These strategies provide flexibility in addressing various video content types, allowing for adaptability in the fusion process. The choice of strategy can be tailored to the specific needs of the task or the nature of the video content being summarized.

---

**Algorithm 1** Calculate Average CLIP Score and Remove Minimum

---

**Require:** summary $s_i \in \mathcal{S}$, video_frames $f_i \in \mathcal{F}$
 1: Initialize empty average CLIP score list $\bar{\mathcal{C}}$
 2: **for** $s_i \in \mathcal{S}$ **do**
 3:     Calculate average CLIP score of $s_i$ with respect to $\mathcal{F}$: $\bar{c} = \frac{1}{|\mathcal{F}|} \sum_{f_i \in \mathcal{F}} \mathrm{CLIP}(s_i, f_i)$
 4:     Store $\bar{c}$ to $\bar{\mathcal{C}}$
 5: **end for**
 6: Locate the index $j$ of the lowest score in $\bar{\mathcal{C}}$
 7: Remove $s_j$ from $\mathcal{S}$.

---

### 3.2 KEYFRAME RETRIEVAL

After obtaining our coherent summary from the Video Summarization module, previous methods either prompt the VideoLLM to generate short segments most relevant to the summary (Qian et al., 2024; Huang et al., 2024), which is an area where current VideoLLMs often underperform, or training a model specifically to encode visual and textual features (Lin et al., 2023b; Moon et al., 2023). The latter approach often employs a sliding window technique to capture and align temporal information, enabling the accurate identification and retrieval of relevant video segments that correspond to the summary. However, this method is computationally expensive and can sometimes result in redundant information.

Given that our textual summary is highly informative, we propose an alternative approach that avoids the need for training a new model. Instead, we utilize a fixed joint embedding space, combined with a similarity metric, to guide the keyframe retrieval. Specifically, we encode the input video frames at two-second intervals, following a sampling technique inspired by Moment-DETR (Lei et al., 2021), alongside our textual summary. Both the text and video frames are encoded using CLIP (Radford et al., 2021). We then calculate the cosine similarity between the text embeddings (whole summary) and the individual frame embeddings, sorting the similarity scores in descending order to identify the top-$k$ video frames as keyframes.

### 3.3 EXTENDED APPLICATIONS

Our method extends beyond existing video summarization, offering practical real-world applications that leverage both our coherent textual summary (from Section 3.1) and retrieved keyframes (from Section 3.2). These include visual manual generation for instructional videos, aiding product manufacturers in creating efficient user guides, and privacy-preserving content generation, which produces short video clips and representative images that capture the essence of the original video without revealing sensitive content. These applications demonstrate our method's versatility, addressing challenges in content creation, information dissemination, and privacy protection across various domains, thus surpassing the capabilities of existing VideoLLMs.

## 4 EXPERIMENTS

### 4.1 EXPERIMENTAL SETUP

Table 1: Statistics of datasets used in our evaluation.

| Dataset | Videos | Video-Qeury Pairs | Avg. Video Len (sec) | Video Types |
|---|---|---|---|---|
| QVHighlights | 10,148 | 10,310 | 150 | Diverse (daily, travel, news, etc.) |
| TACoS | 127 | 18,818 | 287 | Cooking |
| Charades-STA | 9,848 | 18,131 | 30 | Indoor activities |
| DiDeMo | 10,464 | 40,543 | 30 | Diverse (from Flickr) |

**Dataset.**    We evaluate our approach on four well-established datasets: QVHighlights (Lei et al., 2021), TACoS (Regneri et al., 2013), Charades-STA (Gao et al., 2017) and DiDeMo (Anne Hendricks et al., 2017). These datasets span diverse video domains, including sports, product reviews, cooking scenarios, and household activities, etc., providing a comprehensive foundation for assessing our method's performance. Table 1 summarizes the key characteristics of each dataset.

**Implementation Details.**    In our experiments, we employ four VideoLLM "experts": Video-LLaMA (Zhang et al., 2023), Video-LLaVA (Lin et al., 2023a), PG-Video-LLaVA (Munasinghe et al., 2023), and LLaMA-VID (Li et al., 2023b), which serve as both components of our approach and individual baselines. We obtain summaries from each expert, use average CLIP scores to remove outliers, and apply our **Find Common Ground** strategy with Llama-3-8B-Instruct[2] to synthesize the final coherent summary. For keyframe retrieval task, we encode video frames (sampled at two-second intervals) and our generated textual summary into CLIP (Radford et al., 2021) embedding space before calculating similarity metrics.

### 4.2    EXPERIMENTAL RESULTS

**Textual Video Summarization.**    To evaluate the quality of our textual summaries and their alignment with ground truth, we employ G-Eval (Liu et al., 2023a), which we utilize GPT-4-Turbo[3] as the LLM backbone. This method evaluates summaries across seven dimensions: aspect coverage, coherence, faithfulness, fluency, relevance, sentiment consistency, and specificity. Importantly, G-Eval not only assesses video-text alignment through the relevance score but also provides insights into potential human preferences through the remaining metric scores. The results, presented in Table 2, demonstrate that our generated summaries consistently outperform all baseline methods. Our approach achieves superior scores in both video-text alignment and across all aspects that typically correlate with human preference. This comprehensive evaluation underscores the effectiveness of our method in producing high-quality, relevant, and potentially more appealing summaries compared to existing approaches. We also present qualitative results comparing our textual summaries with those of baseline models in Figure 3. While most summaries generated by our baselines capture the essential content, but our approach captures a broader spectrum of information from the given video, providing a more complete and nuanced representation of the content. Also, our method demonstrates potential as an automatic (re-)annotation tool. In cases where ground truth summaries may be inaccurate, as shown in our qualitative results, our framework can serve as a valuable means to verify and potentially correct existing annotations. This capability highlights an additional extensibility of our approach, offering a robust mechanism for enhancing the quality and reliability of video annotation datasets.

**Visual Keyframe Retrieval.**    Following the evaluation metrics in TVR (Lei et al., 2020) and Tall (Regneri et al., 2013), we compute the mean Intersection over Union (mIoU) and Recall@1 with IoU thresholds of 0.5, and 0.7. In addition to individual VideoLLMs as prompt-based baselines, we also include CG-DETR (Moon et al., 2023) as the query-based baseline. The results, presented in Table 3, demonstrate our approach's effectiveness. We outperform all baselines on the Charades-STA, TACoS and DiDeMo datasets, and surpass prompt-based baselines on QVHighlights. Notably, our method, without fine-tuning, achieves superior performance on Charades-STA, TACoS and DiDeMo compared to the fine-tuned CG-DETR. While CG-DETR shows better results on QVHighlights, it's important to consider that CG-DETR benefits from dataset-specific fine-tuning. In contrast, our

---

[2]https://huggingface.co/meta-llama/Meta-Llama-3-8B-Instruct (last accessed: 2024/09)

[3]https://openai.com/index/gpt-4/ (last accessed: 2024/09)

Table 2: Quantitative evaluation of our generated textual video summary among various approaches with G-Eval (Liu et al., 2023a). The best results are marked in **bold**.

| Dimension | OURS | Video-LLaVA | PG-Video-LLaVA | LLaMA-VID | Video-LLaMA |
|---|---|---|---|---|---|
| aspect coverage | **2.77** | 1.31 | 1.85 | 1.97 | 1.72 |
| coherence | **3.35** | 1.56 | 2.12 | 2.76 | 1.83 |
| faithfulness | **2.14** | 1.31 | 1.63 | 1.65 | 1.49 |
| fluency | **3.31** | 1.66 | 2.29 | 2.89 | 2.01 |
| relevance | **2.59** | 1.5 | 1.66 | 1.96 | 1.42 |
| sentiment consistency | **1.92** | 1.23 | 1.38 | 1.6 | 1.31 |
| specificity | **3.22** | 1.41 | 2.12 | 2.44 | 1.97 |

method's strong performance across datasets in a zero-shot setting underscores its robust generalization capabilities. We also provide the qualitative comparison of our keyframe retrieval results against those of our baselines in Figure 4. The visual comparison clearly demonstrates that our selected keyframes achieve a significantly higher coverage rate of the ground truth compared to prompt-based baselines. Moreover, our approach shows superior performance even when compared to CG-DETR. These results visually reinforce the quantitative findings, highlighting our method's effectiveness in accurately identifying and retrieving key moments from videos.

**Extended Applications.** We demonstrate two extended applications of our framework on the HowTo100M (Miech et al., 2019) dataset, which primarily consists of instructional videos. Figure **??** presents the qualitative results of these applications, and more results are provided in the Appendix (cf. A.2). For **visual manual generation**, our generated summary mimic the textual instruction, and the selected keyframes are the visual instructions. This combination of textual and visual elements effectively simulates the creation of visual manuals for instructional content. In the **privacy-preserving content generation**, we utilize Stable Diffusion 3 (Esser et al., 2024) to generate images based on our textual summaries. The resulting images successfully interpret the content of the original videos without revealing sensitive information. These qualitative results illustrate the versatility of our framework in generating practical, real-world applications beyond standard video summarization tasks.

Table 3: Quantitative evaluation of our keyframe retrieval prediction among prompt-based and query-based approaches. The best results are marked in **bold**, and the second-best results are underlined.

| Methods | Charades-STA | | | QVHighlights | | | TACoS | | | DiDeMo | | |
|---|---|---|---|---|---|---|---|---|---|---|---|---|
| | mIoU | R@0.5 | R@0.7 | mIoU | R@0.5 | R@0.7 | mIoU | R@0.5 | R@0.7 | mIoU | R@0.5 | R@0.7 |
| *prompt-based* | | | | | | | | | | | | |
| Video-LLaVA | 0.68 | 0.3 | 0.07 | 9 | 3 | 1.2 | 10.09 | 0.37 | 0 | 6.83 | 2.49 | 0 |
| Video-LLaMA | 5.54 | 1.64 | 0.48 | 5.1 | 1.6 | 0 | 0.13 | 0 | 0 | 6.01 | 0.6 | 0 |
| LLAMA-VID | 20 | 13.79 | 6.73 | 13.8 | 9.3 | 3.6 | 8.23 | 0.34 | 0 | 17.28 | 9.8 | 3.7 |
| PG-Video-LLaVA | 2.57 | 1.32 | 0.57 | 10.1 | 4.4 | 1.3 | 5.7 | 0 | 0 | 7.63 | 1.97 | 0.19 |
| *query-based* | | | | | | | | | | | | |
| CG-DETR | 26.33 | 14.86 | 6.2 | **53.62** | **54.47** | **42.29** | 31.22 | 8.46 | 7.35 | 17.69 | 10.24 | 3.25 |
| Ours | **35.72** | **27.95** | **13.88** | 24.08 | 15.33 | 10.29 | **94.17** | **96.93** | **96.93** | **21.96** | **10.56** | **3.92** |

## 4.3 ABLATION STUDIES

For the experiments in the following studies, the experimental setup follows our main setting in Section 4.1, and we focus on the keyframe retrieval task evaluated on three datasets: QVHighlights, Charades-STA, and TACoS, and metrics: mIoU, R@0.3, R@0.5, and R@0.7, unless otherwise specified.

**Input Video**

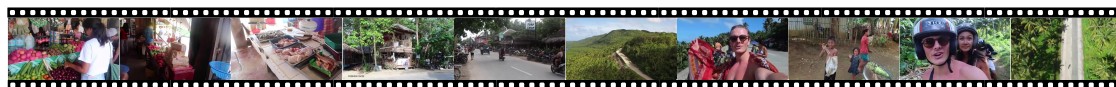

**Textual Summary**

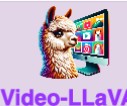

**Video-LLaVA**

This video is about a woman who is seen working at a fruit stand. She is seen arranging the fruits and vegetables on display.

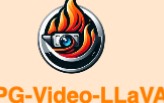

**PG-Video-LLaVA**

The video shows a group of people going on a road trip across an island. They visit a local market and a mud spa, and take photos along the way. The video also shows the group riding scooters and motorcycles, and stopping at a beach resort.

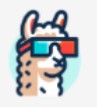

**LLaMA-VID**

The video shows a woman standing in front of a fruit stand, where she is seen picking out fruits. She then walks around the market, and we see her interacting with other people. The video also shows a man standing in front of a buffet, and a woman standing in front of a buffet. We also see a man riding a motorcycle down a road, and a woman standing in front of a buffet. Additionally, we see a man standing in front of a buffet, and a woman standing in front of a buffet. The video ends with a man standing in front of a buffet.

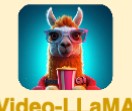

**Video-LLaMA**

This video is about a group of people, including a man and a woman, walking through a market in Indonesia. They are surrounded by various objects such as bananas, chickens, eggs, and bicycles. The video shows the group of people walking through the market and interacting with each other.

**Ground Truth**

Man is wearing shirt around one shoulder on the side of the road.

**Ours**

The video showcases a group of people exploring an island, visiting a local market, and engaging in various activities. The group is seen interacting with each other and with the locals, and they also take photos along the way. The video features scenes of people riding scooters and motorcycles, and stopping at a beach resort. Additionally, the group is seen walking through a market, surrounded by various objects such as fruits, vegetables, and bicycles, and interacting with vendors and other people.

Figure 3: Visualization of textual video summaries generated by individual VideoLLMs and our proposed collaboration approach. Keyframes are displayed at the top as input video. Additionally, we provide the ground truth summary for reference.

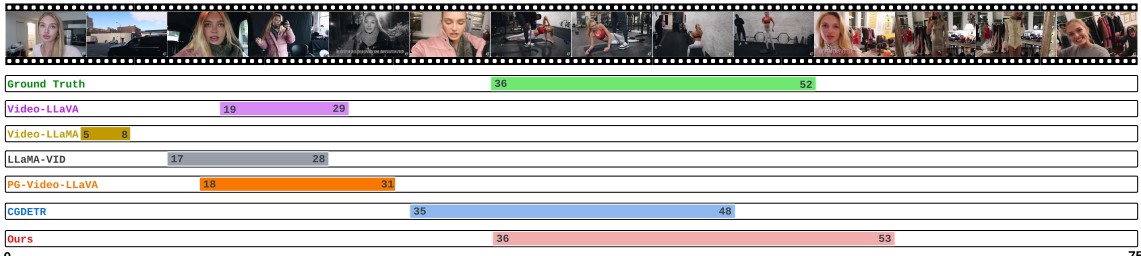

Figure 4: Visualization of prediction results comparison on QVHighlights (Lei et al., 2021). The ground truth keyframes are shown at the top as the input video, and the prediction unit is in seconds.

Table 4: Ablation study of the effect of filtering outliers.

| | QVHighlights | | | | Charades-STA | | | | TACoS | | | |
|---|---|---|---|---|---|---|---|---|---|---|---|---|
| | mIoU | R@0.3 | R@0.5 | R@0.7 | mIoU | R@0.3 | R@0.5 | R@0.7 | mIoU | R@0.3 | R@0.5 | R@0.7 |
| w Filter outliers | 23.93 | 21.93 | 14.88 | 10.29 | 35.16 | 36.1 | 27.03 | 13.94 | 94.37 | 97.04 | 97.04 | 97.04 |
| w/o Filter outliers | 21.92 | 19.53 | 12.9 | 7.35 | N/A | N/A | N/A | N/A | 92.27 | 95.89 | 95.89 | 95.89 |

Table 5: **Ablation study of effects of different cooperation strategies utilizing different LLMs.** In the "LLM" column, "GPT" represents "GPT-4-Turbo", and "LLaMA" denotes "LLaMA-3-8b-Instruct". In the "Cooperate Strategy" column, "CG" refers to the "Find Common Ground" strategy, while "M" stands for the "Merge" strategy. The best results are marked in **bold**.

| LLM | Cooperate Strategy | QVHighlights | | | | Charades-STA | | | | TACoS | | | |
|---|---|---|---|---|---|---|---|---|---|---|---|---|---|
| | | mIoU | R@0.3 | R@0.5 | R@0.7 | mIoU | R@0.3 | R@0.5 | R@0.7 | mIoU | R@0.3 | R@0.5 | R@0.7 |
| GPT | CG | 23.93 | 21.93 | 14.88 | **10.29** | 35.16 | 36.1 | 27.03 | 13.94 | 94.37 | 97.04 | 97.04 | 97.04 |
| | M | 23.45 | 21.23 | 14.13 | 9.81 | **35.81** | 37.3 | 27.82 | **14** | **94.4** | **97.06** | **97.06** | **97.06** |
| LLaMA | CG | **24.08** | **22.31** | **15.33** | **10.29** | 35.41 | 37.33 | 27.27 | 13.72 | 94.05 | 96.86 | 96.86 | 96.31 |
| | M | 23.1 | 20.84 | 13.92 | 9.43 | 35.72 | **37.76** | **27.95** | 13.88 | 94.18 | 96.93 | 96.93 | 96.93 |

**Effect of filtering outliers.** To assess the impact of our "Filter Outliers" component, we compare our framework's performance with and without this feature. In both scenarios, we utilize GPT-4-Turbo to synthesize summaries from individual VideoLLMs using the "Find Common Ground" strategy. The key difference lies in the input to this fusion process: with outlier filtering, we exclude the detected outlier, while without it, all four summaries are included. As demonstrated in Table 4, the inclusion of outlier filtering led to a significant improvement in performance, enhancing both mean Intersection over Union (mIoU) and Recall metrics by at least 2%. This consistent improvement across metrics underscores the effectiveness of our outlier filtering approach in refining the quality of the final summary.

**Effect of different cooperation strategies with different LLMs.** We examine the impact of different cooperation strategies and LLMs on our framework's performance. We compare two cooperation strategies, Merge and Find Common Ground, implemented with two distinct LLMs: the open-source Llama-3-8b-Instruct and the closed-source GPT-4-Turbo, and the prompt template we apply is provided in the Appendix (cf. A.1). We present the results in Table 5. Our analysis reveals that the choice of LLM and cooperation strategy has only marginal effects on the overall performance. However, all combinations demonstrate substantial improvements over utilizing only the individual VideoLLM summaries, as shown in Table 3. Our results strongly suggest that our method of combining and refining summaries from multiple VideoLLMs produces more comprehensive and accurate textual representations, which in turn lead to improved keyframe selection.

**Effect of audio information.** To assess the influence of audio information, we conduct experiments with and without audio input, noting that some VideoLLMs, such as Video-LLaMA, incorporate audio information, others like Video-LLaVA and LLaMA-VID do not include this modality in their frameworks. For Video-LLaMA, we remove the audio branch to simulate scenarios without audio information. In the case of PG-Video-LLaVA, we deactivate the audio branch in our default setting. The results, presented in Table 6, demonstrate the significant contribution of audio information to the quality of video summaries. Including audio led to a 5-10% improvement in downstream keyframe retrieval performance.

Table 6: **Ablation study of the impact of audio information.** We remove the audio branch of Video-LLaMA (Zhang et al., 2023) to simulate the case of "w/o audio".

| | QVHighlights | | | | Charades-STA | | | |
|---|---|---|---|---|---|---|---|---|
| | mIoU | R@0.3 | R@0.5 | R@0.7 | mIoU | R@0.3 | R@0.5 | R@0.7 |
| w audio | 24.08 | 22.31 | 15.33 | 10.29 | 35.16 | 36.1 | 27.03 | 13.94 |
| w/o audio | 19.45 | 17.28 | 10.73 | 7.1 | 27.75 | 22.24 | 10.58 | 2.5 |

## 5 CONCLUSION

We propose a holistic video summarization framework that leverages multiple VideoLLMs to generate comprehensive textual summaries that capture the detail of the given video without fine-tuning. Our extensive experiments demonstrate the effectiveness of our method in downstream tasks like keyframe retrieval and extended applications such as visual manual generation and privacy-preserving content creation. Our framework's adaptability allows for easy integration of more advanced models, ensuring its relevance as the field progresses. By establishing a foundation for integrating visual and linguistic information, our approach paves the way for more sophisticated multimedia analysis tools. We anticipate that this framework will catalyze advancements in video understanding and natural language processing, leading to more intuitive and powerful systems across various domains.

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

# A  APPENDIX

## A.1  MERGING PROMPT

**System Prompt:**
As an AI specializing in video summarization, your task is to analyze and find the common ground from following paragraphs of video summaries. Which the common ground truth means the similar description appears in each four paragraphs.These summaries are generated from multiple video understanding models, all of which processed the same input videos.

**User Prompt:**
Find the common ground of the following paragraphs and make it into a coherent paragraph:
Content:
1. {VideoLLaVA}
2. {PGVideoLLaVA}
3. {LLaMAVID}
4. {VideoLLaMA}

Figure 5: Prompt template of "Find common ground" strategy in the cooperation step.

**System Prompt:**
As a video summarization expert, your purpose is to combine and summarize multiple paragraphs of summary generated from different video understanding models. You will take the summaries provided as input and transform them into a smooth and coherent paragraph. Additionally, you will automatically discard any irrelevant parts to ensure the final summary is concise and relevant. With your expertise in video summarization, you will help me extract the most important information from the given summaries and present it in a comprehensive manner.

**User Prompt:**
Combine the following four paragraphs into a cohesive, single paragraph while maintaining the overall essence and information provided by each.
Content:
1. {VideoLLaVA}
2. {PGVideoLLaVA}
3. {LLaMAVID}
4. {VideoLLaMA}

Figure 6: Prompt template of "Merge" strategy in the cooperation step.

## A.2  HOWTO100M QUALITATIVE RESULT

## A.3  CHARADES-STA QUALITATIVE RESULT

**Input Video**

**Textual Summary**

The video is an instructional video that shows how to use a rope to pull a tree down. The video starts with a man holding a rope and a box of tools. He then shows how to use the tools to pull the tree down. The video also shows how to use a rope to tie the tree down. The man then shows how to use a rope to pull the tree down.

**Genarative Images by Stable Diffusion**

**Input Video**

**Textual Summary**

The video is an instructional video that shows a man installing a door trim. The man is using a tool to cut the trim, and he is also using a drill to attach the trim to the door. The video also shows the man using a tape measure to measure the trim and a sandpaper to smooth out the edges. The video ends with the man finishing the installation of the door trim.

**Genarative Images by Stable Diffusion**

**Input Video**

**Textual Summary**

The video shows a man working on a car in a garage. He is using a laptop to connect to the car's computer system and diagnose the issue. The man then explains the problem and how to fix it.

**Genarative Images by Stable Diffusion**

Figure 7: HowTo100M textual summary, selected keyframes, and generative results.

**Input Video**

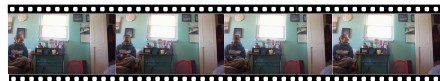

**Textual Summary**

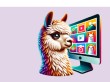

**Video-LLaVA**

This video is about a girl who is sitting on a bed and talking to the camera. She then proceeds to put on a pair of socks and a shirt.

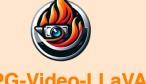

**PG-Video-LLaVA**

The video shows a woman sitting in a room, holding a cup and a blue towel. She is wearing a blue sweater and appears to be drinking from the cup. The woman is also holding a blue towel in her hand. The video does not provide any additional information about the content of the video.

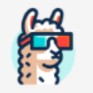

**LLaMA-VID**

The video features a woman who is sitting on a couch and talking to the camera. She is holding a cup and a remote control in her hands. The woman takes a sip of water from the cup and then continues to talk to the camera. The video seems to be a casual conversation between the woman and the camera, with no apparent context or purpose. The woman appears to be comfortable and relaxed in her surroundings, and her body language suggests that she is at ease in front of the camera. The lighting in the video is soft and natural, with no artificial light sources visible. Overall, the video seems to be a simple and intimate moment captured on camera.

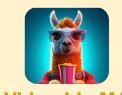

**Video-LLaMA**

This video is about a young woman sitting on a couch in a bedroom, holding a vacuum cleaner and looking at her phone. She is wearing a blue jacket and has long brown hair. There is also a black bag on the couch next to her.

**Ground Truth**

A person is playing on the bed and laughing at the doorway.

**Ours**

In the video, a young woman with a ponytail and glasses is depicted twice, once focusing on her sitting on a bed surrounded by an array of objects such as a clock, cup, bottle, vase, book, and remote, and another time showing her smiling and engaging with her cellphone in a room characterized by a blue wall and similarly filled with items including a mirror, bookshelf, and a picture. Additionally, a separate scene captures another woman with long, dark hair, standing in front of a vanity mirror in a well-lit room brushing her hair, dressed casually without any visible distractions, emphasizing a serene moment of personal grooming. The settings are simple, with natural light enhancing the intimate atmospheres of the scenes.

Figure 8: Charades-STA textual summary.

