# OpenReview forum: "Realizing Video Summarization from the Path of Language-based Semantic Understanding"
_ICLR.cc/2025/Conference — ICLR 2025 Conference Withdrawn Submission_

### Official Review · Reviewer_PbC7 · 2024-10-30

**Soundness:** 3
**Presentation:** 3
**Contribution:** 2
**Rating:** 3
**Confidence:** 4

**Summary:**

This submission focuses on the task of video summarization. Specifically, it integrates multiple VideoLLMs to generate comprehensive and cohrent textual summaries without additional fine-tuning. The proposed model also enhance the quality of downstream tasks that required rich semantic extraction and retrieval capabilities.

**Strengths:**

+ This paper propose a Mixture of Experts inspired paradigm that jointly utilising multiple off-the-shelf VideoLLMs via LLM cooperation. The resulting system can be easily upgraded by incorporating more VideoLLMs.
+ This paper proposes an inference-time algorithm to generate textual summaries, key frames identification, and visual summary generation.

**Weaknesses:**

- The first strategy in filter outliers is based on the assumption that most of the LLM will output correct and similar. The second strategy compare visual and textual embedding for correctness of the summary. While these are reasonable for simpler video, it may face problems if the input video is rich and highly diverse, and exists multiple events. The definition of "good summary" is highly depend on the quality of the selected VideoLLMs.

**Questions:**

The paper is well written and easy to follow. I believe it is a great engineering project that shows how off-the-shelf VideoLLMs can be leveraged with a inference-time algorithm to generate a single textual summaries. My main concern is the technical novelty of this submission. It appear to me that the paper only provides a way to combined existing models to produce a better summary. In other words, the results are benefitted mainly from the capability of pre-trained LLM and VideoLLMs. I would really like the authors to outline what is the novel technical contributions and/or insights generated from this approach that is beneficial to researcher in this field.

---

### Official Review · Reviewer_r1cG · 2024-11-03

**Soundness:** 3
**Presentation:** 1
**Contribution:** 1
**Rating:** 1
**Confidence:** 5

**Summary:**

This paper presents a text summary generation method for videos by combining multiple fundamental models as a mixture of experts. The easy-to-understand flow is presented in Figure 2. A summary pool is generated by using multiple VideoLLMs. Then, the summaries are polished (outliers are filtered out) to generate a better summary. Some applications such as key frame extraction and abstracted key story generation.

**Strengths:**

+ The quantitative evaluation for text summary generation and keyframe prediction shows superior performance as compared to the subset of the proposed method.
+ Some possible applications are demonstrated, which might be useful.

**Weaknesses:**

- Technical novelty is very weak.
  - As can be seen in Section 3, this work is just a combination of existing fundamental models. The flow and how to combine them might be the main scope of this paper, but I could not see any novel technical contributions in this paper. The authors might want to clarify their technical novelty.
  - No technical details are given in the manuscript. Some key components are presented such as “filter outliers” and “cooperate,” but their technical details are all missing. The authors might want to show the equations or the pseudo code of them so that possible readers can grab the idea. It is almost impossible to reproduce the proposed method.

- Experimental comparison is not good enough
  - The proposed work is compared only with the subset of their proposed model: each LLM without MoE. It is apparent that the mixture model would outperform a single model. The author might want to design the experimental comparison better.
  - In Section 4, only the numerical comparison is discussed. Since the task is the video summarization, the authors should show the generated summaries and discuss how they are qualitatively different (I see a set of examples in Figure 3, but this is just an example, no detailed discussion is given).
  - In addition, the authors might want to discuss the failure cases and limitations.

Here are some comments to improve the paper:
- reference error in line 346

**Questions:**

Please answer my comments in the Weakness part.

---

### Official Review · Reviewer_R7eR · 2024-11-03

**Soundness:** 2
**Presentation:** 2
**Contribution:** 1
**Rating:** 3
**Confidence:** 4

**Summary:**

The paper presents a method for textual video summarization using multiple multimodal LLMs. The multiple multimodal LLMs individually takes video frames and generates textual summaries, which are then reduced to a single summary for the input video. The method is evaluated on several datasets, showing performance gains. Ablation studies are also provided. Some applications are also shown,

**Strengths:**

The method improves the performance. Some application-oriented textual video summarization may benefit from the paper. Also, the applications of the method are interesting.

**Weaknesses:**

The idea of the paper, which is to use multiple LLMs to generate multiple video summaries and then integrate them into a single video summary, looks naïve, and people can expect that they perhaps improve the performance. The technical contribution of the paper looks weak, and no interesting insights are provided. I would like to see more specific descriptions about why these results are worth sharing with the community.

The first point of the summary of the contributions, saying the method an generate single coherent, and unbiased summary, does not make much sense for me. What does it mean by a coherent and unbiased summary? As far as I understand, the experimental results do not say anything related to coherency and unbiasedness of generated summaries.

**Questions:**

I would like to see the responses to both of two points raised as the weaknesses.

---

### Official Review · Reviewer_iKgU · 2024-11-03

**Soundness:** 3
**Presentation:** 2
**Contribution:** 2
**Rating:** 3
**Confidence:** 5

**Summary:**

* This work introduces a new framework inspired by the Mixture of Experts (MoE) approach, allowing VideoLLMs to complement each other’s strengths without fine-tuning.
* It provides detailed background descriptions and identifies keyframes, improving semantically meaningful video retrieval over traditional methods that rely solely on visual features.
* The summaries enhance performance in downstream tasks such as summary video generation, both through keyframe selection and by using text-to-image models.

**Strengths:**

* Unlike other models that require resource-intensive fine-tuning, this framework works at inference time, saving computational resources and time and showing descent performance in various downstream tasks.

* The framework is flexible and can incorporate new VideoLLMs as they are developed, allowing it to stay current with advancements in the field.

**Weaknesses:**

1. The main weakness of this work lies in the novelty part. In addition, this work is more suitable for submission to application-related conferences.

   (a). The filtering strategy is more like an ensemble method that tries to get the consensus agreement across models via similarity matching, which is a common approach.

   (b). For the keyframe retrieval part, utilizing CLIP via similarity score across image and text is already well explored in [1,2].

   (c). The high-level idea of summarizing video using VLM and LLM was explored in [3], which should also be cited.

   (d). The idea of visualizing the visual steps for summarization is interesting. However, it was also explored in [4], where it generate the instruction step by diffusion models.

2. In the experiment, the proposed model in Table 3 is significantly higher than the baseline with 94.17, which is nearly identical to ground truth. Is there any reason behind this? The result seemed seemed abnormal.

3. The paper requires proofreading polished L346 ?? presents. Table 3 is out of bound.

4. The framework relies on the availability and compatibility of multiple VideoLLMs, which may not always be feasible or efficient for every application, particularly in low-resource environments.

[1] Luo, Huaishao, et al. "Clip4clip: An empirical study of clip for end-to-end video clip retrieval and captioning." Neurocomputing 508 (2022): 293-304.

[2] Portillo-Quintero, Jesús Andrés, José Carlos Ortiz-Bayliss, and Hugo Terashima-Marín. "A straightforward framework for video retrieval using clip." Mexican Conference on Pattern Recognition. Cham: Springer International Publishing, 2021.

[3] Argaw, Dawit Mureja, et al. "Scaling Up Video Summarization Pretraining with Large Language Models." Proceedings of the IEEE/CVF Conference on Computer Vision and Pattern Recognition. 2024.

[4] Souček, Tomáš, et al. "Genhowto: Learning to generate actions and state transformations from instructional videos." 2024 IEEE/CVF Conference on Computer Vision and Pattern Recognition (CVPR). IEEE, 2024.

**Questions:**

1. Please address the weakness 1,2,4.
2. How were the generated image frames evaluated?

---

### Note · Authors · 2024-11-15

I have read and agree with the venue's withdrawal policy on behalf of myself and my co-authors.